# UNO: Unlearning via Orthogonalization in Generative Models

## Abstract

As generative models become increasingly powerful and pervasive, the ability to unlearn specific data, whether due to privacy concerns, legal requirements, or the correction of harmful content, has become increasingly important. Unlike in conventional training, where data are accumulated and knowledge is reinforced, unlearning aims to selectively remove the influence of particular data points without costly retraining from scratch. To be effective and reliable, such algorithms need to achieve (i) forgetting of the undesired data, (ii) preservation of the quality of the generation, (iii) preservation of the influence of the desired training data on the model parameters, and (iv) small number of training steps. We propose fast unlearning algorithms based on loss gradient orthogonalization for unconditional and conditional generative models. We show that our algorithms are able to forget data while maintaining the fidelity of the original model. On standard image benchmarks, our algorithms achieve orders of magnitude faster unlearning times than their predecessors, such as gradient surgery. We demonstrate our algorithms with datasets of increasing complexity (MNIST, CelebA and ImageNet-1K) and for generative models of increasing complexity (VAEs and diffusion transformers).

## 1 Introduction

Machine learning models are often trained on datasets that contain personal or sensitive information, such as medical records, financial data, or social media activity (Mireshghallah et al., 2021; Truong et al., 2021). This reliance on personal data introduces substantial privacy risks, especially when models can unintentionally memorize or leak identifiable information; see Carlini et al. (2021) for an in-depth exploration of this issue in the context of large language models (LLMs). Legal frameworks such as the General Data Protection Regulation (GDPR) and related EU laws have been established to address these issues (gdp, 2016). One of the central provisions is the right to be forgotten (RTBF), which grants individuals the ability to request the deletion of their personal data (Kuner et al., 2020). It is increasingly likely that this obligation will become a standard requirement for machine learning services. Retraining large models from scratch each time such a request is received is computationally infeasible since the training costs are substantial (Brown et al., 2020; Hoffmann et al., 2022). Machine unlearning refers to the removal of the influence of specific data points from a trained model without requiring full retraining. In the context of generative models this can be formalized as follows. Given a training dataset $\mathcal{D} = \mathcal{D}_r \sqcup \mathcal{D}_f$ partitioned into retain and forget datasets $\mathcal{D}_r$ and $\mathcal{D}_f$, respectively, and a model $\mathcal{M}_\theta$ trained on $\mathcal{D}$, the objective of unlearning is to update the model parameters $\theta$ in a way such that $P(\mathrm{sim}(x, \mathcal{D}_f) \geq \delta) \leq \varepsilon$ where $P$ denotes probability, $x$ is a sample generated by the updated model, $\mathrm{sim}$ is an appropriate similarity measure, and $\varepsilon, \delta$ are thresholds controlling the degree of forgetting. For an unlearning algorithm to be effective, it should (i) prevent the model from generating data resembling samples from $\mathcal{D}_f$, (ii) preserve the quality or fidelity of the generated samples, (iii) retain the influence of $\mathcal{D}_r$ on the model parameters, and (iv) require only a small number of training steps.

A simple approach to machine unlearning is to reverse the model update steps by performing gradient ascent on the loss computed over the forget dataset $\mathcal{D}_f$. However, this method is susceptible to catastrophic forgetting, where the model loses knowledge far beyond just the targeted forget dataset (McCloskey & Cohen, 1989; Luo et al., 2023). To mitigate this, several approaches combine gradient ascent on $\mathcal{D}_f$ with gradient descent on the retain dataset $\mathcal{D}_r$ (Yao et al., 2024). The Gradient Difference (GDiff) method minimizes the difference of losses evaluated on the retain and forget

datasets. Balancing the opposing updates in ascent-descent methods or weighing the loss terms properly in methods like GDiff is challenging since the forget and retain datasets might have significant size disparity, and the risk of catastrophic forgetting persists unless training hyperparameters such as the learning rate are finely tuned (Bu et al., 2024). Recently, multi-task optimization (MTO) techniques have inspired several unlearning algorithms (Sener & Koltun, 2018; Yu et al., 2020). One such algorithm is gradient surgery (Bae et al., 2023) where gradient ascent is performed in a direction that is orthogonal to the loss gradient computed over the retain dataset. This method, however, remains sensitive to the choice of hyperparameters and can suffer from catastrophic forgetting without careful tuning, see Appendix D for an example.

In this work, we aim to advance the gradient surgery framework for unlearning in generative models. Although our proposed algorithms are general-purpose and presented accordingly, we demonstrate their effectiveness specifically using variational autoencoders (VAEs) (Kingma & Welling, 2013; Rezende et al., 2014) and diffusion transformers (Peebles & Xie, 2023) for three widely used benchmark image datasets MNIST (Deng, 2012), CelebA (Liu et al., 2015) and ImageNet-1K (Deng et al., 2009).

CONTRIBUTIONS

Our main contributions are as follows:

1. We propose two new unlearning algorithms that regularize the main loss function with an additional term enforcing orthogonality between loss gradients computed over the retain and forget datasets. We provide algorithms for both unconditional and conditional generative models.

2. We compare our algorithms against prior approaches, including gradient surgery and gradient ascent, evaluating both unlearning speed and the quality of generated samples. Our methods achieve orders of magnitude faster unlearning than gradient surgery, while retaining the influence of the desired training data unlike gradient ascent.

3. We provide implementations of both the proposed and baseline algorithms, along with the experiment data, through this anonymized file.

## 2 RELATED WORK

Early foundational work by Koh and Liang Koh & Liang (2017) introduced influence functions as a principled approach for quantifying the impact of removing individual training points from machine learning models. Although influential, their technique is computationally demanding, limiting its scalability, particularly for large-scale neural networks (Basu et al., 2020; Guo et al., 2020). To address these computational challenges, recent studies have developed more efficient and scalable methodologies. For example, Schioppa et al. (2022) and Guo et al. (2020) proposed efficient approximations of influence functions that significantly reduce computational complexity. Further, innovative optimization-based frameworks such as SCRUB by Kurmanji et al. (2023) approximate data removal for classification models such as ResNet (He et al., 2016) using a teacher-student distillation paradigm combined with checkpoint rewinding. Gradient-based methods have emerged as an effective paradigm for machine unlearning. Golatkar et al. (2020) approximate the influence of individual data points on model parameters using the Fisher Information Matrix and use it to execute unlearning in deep networks. Building on this, Mixed-Privacy Forgetting (Golatkar et al., 2021) combines public and private data during training, enabling the selective removal of private data while preserving the utility of public data. Neel et al. (2021) propose Descent-to-Delete, a gradient-based optimization technique that incrementally updates model parameters to approximate the behavior of a model trained without the forgotten data.

Unlearning in generative models introduces distinct challenges due to their capacity to implicitly memorize training data, complicating data removal without degrading generative quality. Addressing these, Sun et al. (2025) introduced methods specifically designed to detect and mitigate unintended memorization in generative adversarial networks (GANs). Heng & Soh (2023) proposed Selective Amnesia, which leverages continual learning frameworks to selectively remove specific concepts from deep generative models without compromising the overall data distribution learned by the model.

In the context of LLMs, recent works have tackled critical challenges such as selective forgetting of harmful or copyrighted content and aligning models to user preferences (Jang et al., 2022; Chen & Yang, 2023; Qu et al., 2025; Pawelczyk et al., 2023). These methods employ parameter-efficient fine-tuning, low-rank adaptations, and in-context learning strategies to remove specific learned knowledge while minimally impacting overall model performance. Style unlearning in the context of text-to-image models has been recently studied, using negative classifier-free guidance (Gandikota et al., 2023).

Negative Preference Optimization (NPO) (Zhang et al., 2024) offers an alignment-inspired approach to machine unlearning by assigning lower preference or likelihood to data from the forget set. Through preference-based training, the model learns to reduce its reliance on forget data, often using pairwise comparisons or preference signals. Normalized Gradient Difference (NGDiff) (Bu et al., 2024) approaches unlearning as a multi-task optimization problem, balancing the objectives of forgetting and retaining. By normalizing the gradient differences between these tasks and employing an adaptive learning rate scheduler, NGDiff provides stable training and effectively manages the trade-off between unlearning and model utility. Cao et al. (2022) propose a projection residual based method to remove the influence of undesired data. In the same vein, gradient surgery (Bae et al., 2023) attempts to maximize the loss in a direction orthogonal to the loss gradient evaluated on the retain dataset. While promising for generative models, gradient surgery can suffer from inefficiency when there is significant overlap between loss gradients computed on the retain and forget data, and may even cause catastrophic forgetting. We aim to improve upon this approach by explicitly enforcing orthogonality between these conflicting gradients. Our algorithms exhibit no catastrophic forgetting, achieve fast unlearning speeds, and are robust to hyperparameter selection.

For a comprehensive overview of unlearning techniques for large language models, including method categorization and scale-specific challenges, see Blanco-Justicia et al. (2025). For a broad taxonomy of machine unlearning across centralized, distributed, and privacy-critical settings with a focus on open problems and verification, see Wang et al. (2024).

## 3 UNLEARNING VIA ORTHOGONALIZATION

We now describe the unlearning algorithms used to produce the results presented in this paper. The pseudocode for all the algorithms presented in this section can be found in Appendix B. We first describe classical gradient ascent and gradient surgery before introducing our Unlearning via Orthogonalization (UNO) algorithms with and without surgery.

### 3.1 GRADIENT ASCENT

We begin by introducing the most primitive approach, namely, gradient ascent. Given a pretrained model $\mathcal{M}_\theta$ with trained parameters $\theta = \theta^\star$, trained using a loss function $\mathcal{L}$ on a dataset $\mathcal{D}$, unlearning can be induced by maximizing the loss on the forget data, which can be done with the update step:

$$\theta_{k+1} = \theta_k + \eta \mathbf{g_f}, \tag{A}$$

where $\theta_k$ represents the model parameters after the $k$-th training step with $\theta_0 = \theta^\star$, $\eta$ is the learning rate, and $\mathbf{g_f}$ is the gradient of the loss evaluated over the forget data (we omit the index of $\theta$ in the definition below for brevity),

$$\mathbf{g_f} = \frac{1}{|\mathcal{D}_f|} \sum_{x \in \mathcal{D}_f} \nabla_\theta \mathcal{L}(\mathcal{M}_\theta, x). \tag{1}$$

This approach, however, may delete knowledge acquired on the retain data $\mathcal{D}_r$ if $\mathbf{g_f}$ resembles $\mathbf{g_r}$, the gradient of loss evaluated over the retain data,

$$\mathbf{g_r} = \frac{1}{|\mathcal{D}_r|} \sum_{x \in \mathcal{D}_r} \nabla_\theta \mathcal{L}(\mathcal{M}_\theta, x). \tag{2}$$

A naive way to prevent the model from forgetting retain data is to perform alternating ascent in the direction of $\mathbf{g_f}$ and descent in the direction of $\mathbf{g_r}$:

$$\theta_{k+1} = \begin{cases} \theta_k + \eta \mathbf{g_f}, & \text{if } k \text{ is even}, \\ \theta_k - \eta \mathbf{g_r}, & \text{if } k \text{ is odd}. \end{cases} \tag{A-D}$$

This simple modification, which we will refer to as ascent-descent, does not safeguard against catastrophic forgetting, as we will see in Section 4. See Appendix C for examples of catastrophic forgetting induced by gradient ascent and ascent-descent.

## 3.2 GRADIENT SURGERY

Similar challenges also arise in a related subfield of machine learning: multi-task optimization where a model must learn to perform new tasks without compromising performance on earlier tasks (Crawshaw, 2020). If the loss gradient corresponding to the new task points in a direction opposing the loss gradients corresponding to the old tasks, the model risks losing its previously learned skills with each new gradient descent step, paralleling catastrophic forgetting. In multi-task optimization, gradient surgery refers to techniques that modify task-specific gradients during training to reduce this interference between tasks. When gradients from different tasks conflict, i.e., point in opposing directions, methods like PCGrad project gradients to minimize this conflict, allowing the model to learn multiple tasks more effectively without one task hindering the progress of another (Yu et al., 2020).

Gradient surgery can be used to reduce the potential conflict between $\mathbf{g_f}$ and $\mathbf{g_r}$ to improve the vanilla gradient ascent (Bae et al., 2023) via removing the orthogonal projection of $\mathbf{g_r}$ from $\mathbf{g_f}$ before taking the ascent step:

$$\bar{\mathbf{g}}_{\mathbf{f}} = \mathbf{g_f} - \frac{\mathbf{g_r} \cdot \mathbf{g_f}}{\mathbf{g_r} \cdot \mathbf{g_r}} \mathbf{g_r}, \tag{SA}$$
$$\theta_{k+1} = \theta_k + \eta \bar{\mathbf{g}}_{\mathbf{f}}.$$

While this modified ascent reduces over-unlearning compared to vanilla ascent, it does not fully resolve the issue, and still requires careful tuning of $\eta$ to avoid catastrophic forgetting. Therefore, we introduce another version of gradient surgery which we find to be more stable and use it throughout, for generating the results in Section 4. Rather than perform ascent along modified $\mathbf{g_f}$ direction, we perform descent along modified $\mathbf{g_r}$ direction resulting in the following update:

$$\bar{\mathbf{g}}_{\mathbf{r}} = \mathbf{g_r} - \frac{\mathbf{g_r} \cdot \mathbf{g_f}}{\mathbf{g_f} \cdot \mathbf{g_f}} \mathbf{g_f}, \tag{S}$$
$$\theta_{k+1} = \theta_k - \eta \bar{\mathbf{g}}_{\mathbf{r}},$$

which aims at minimizing the loss in directions orthogonal to $\mathbf{g_f}$. This form of gradient surgery does not suffer from catastrophic forgetting, is robust to the choice of $\eta$, and consequently can achieve faster unlearning speeds compared to (SA) with larger values of $\eta$. For a comparison of these two versions of gradient surgery: (SA) and (S), see Appendix D.

## 3.3 UNO AND UNO-S

In the ideal scenario, when $\mathbf{g_f}$ is orthogonal to $\mathbf{g_r}$, (SA) is equivalent to gradient ascent (A) without the risk of losing desired knowledge. Furthermore, (S) is equivalent to retraining the model on the retain data, without the risk of relearning about the forget data. Therefore, we propose a modified loss function that attempts to enforce this ideal scenario with the help of an orthogonality promoting regularization term,

$$\mathcal{L}_{\text{UNO}} = \frac{1}{|\mathcal{D}_r|} \sum_{x \in \mathcal{D}_r} \mathcal{L}(\mathcal{M}_\theta, x) + \beta_o \left( \frac{\mathbf{g_r} \cdot \mathbf{g_f}}{\|\mathbf{g_r}\| \|\mathbf{g_f}\|} \right)^2, \tag{3}$$

where $\beta_o$ is a regularization parameter. The unlearning via orthogonalization algorithm (UNO), can be expressed as performing gradient descent on this modified loss,

$$\theta_{k+1} = \theta_k - \eta \nabla_{\theta_k} \mathcal{L}_{\text{UNO}}. \tag{UNO}$$

Note that we only use the retain data to construct the first term in (3) to mimic the ideal retraining scenario mentioned above.

We further propose a hybrid algorithm that applies the (UNO) update step and the (S) update step alternately which we refer to as UNO-S:

$$\theta_{k+1} = \begin{cases} \theta_k - \eta \nabla_{\theta_k} \mathcal{L}_{\text{UNO}}, & \text{if } k \text{ is even,} \\ \theta_k - \eta \bar{\mathbf{g}}_{\mathbf{r}}, & \text{if } k \text{ is odd.} \end{cases} \tag{UNO-S}$$

The UNO update step attempts to enforce orthogonality between $\mathbf{g_f}$ and $\mathbf{g_r}$, which helps the subsequent surgery step to effectively resolve the conflict between them.

## 3.4 Replacement unlearning for conditional generative models

In the case of conditional generation, we can aim to replace the generation corresponding to an undesired condition $c_f$ with the generation corresponding to a target condition $c_t$. In this scenario it is natural to minimize the following quantity,

$$\mathcal{L}^R = \frac{1}{|\mathcal{D}_t|} \sum_{x \in \mathcal{D}_t} \mathcal{L}(\mathcal{M}_\theta(c_f), x) + \frac{1}{|\mathcal{D}_t|} \sum_{x \in \mathcal{D}_t} \mathcal{L}(\mathcal{M}_\theta(c_t), x), \tag{4}$$

where $\mathcal{D}_t$ is the data corresponding to $c_t$. The first term enforces replacement and the second term represents the conditional variant of the retain loss that appears in (2). Using $\mathcal{L}^R$, we can devise the conditional variants of (S), (UNO) and (UNO-S) by executing (UNO) with

$$\mathcal{L}_{\mathrm{UNO}}^R = \mathcal{L}^R + \beta_o \left( \frac{\mathbf{g_r^R} \cdot \mathbf{g_f^R}}{\|\mathbf{g_r^R}\|\|\mathbf{g_f^R}\|} \right)^2, \tag{5}$$

where

$$\mathbf{g_r^R} = \nabla_\theta \mathcal{L}^R, \tag{6}$$

$$\mathbf{g_f^R} = \frac{1}{|\mathcal{D}_f|} \sum_{x \in \mathcal{D}_f} \nabla_\theta \mathcal{L}(\mathcal{M}_\theta(c_f), x). \tag{7}$$

Here $\mathcal{D}_f$ again denotes the forget data associated with the condition $c_f$. The surgery steps (S) and (UNO-S) use

$$\bar{\mathbf{g}}_{\mathbf{r}}^{\mathbf{R}} = \mathbf{g_r^R} - \frac{\mathbf{g_r^R} \cdot \mathbf{g_f^R}}{\mathbf{g_f^R} \cdot \mathbf{g_f^R}} \mathbf{g_f^R}. \tag{8}$$

## 3.5 Classifier-assisted unlearning

We further consider the case when we may have access to a binary classifier that distinguishes forget data from retain data and we can leverage this extra information to accelerate unlearning algorithms. We can use this classifier to identify every sample generated by our model as either a retain or forget sample, and compute the probability $p_r$ that a generated sample is a retain sample. This associates our generative model with a Bernoulli distribution with probability of success $p_r$. We would like this distribution to have probability of success close to 1 with $1 - \alpha$ where $\alpha$ is a small positive threshold controlling the degree of forgetting. We can enforce this by simply adding the following term to our loss,

$$\beta_h d_{\mathrm{KL}} = \beta_h \left[ p_r \log \left( \frac{p_r}{1 - \alpha} \right) + (1 - p_r) \log \left( \frac{1 - p_r}{\alpha} \right) \right], \tag{9}$$

where $\beta_h$ is a regularization parameter, and $d_{\mathrm{KL}}$ represents the KL divergence between the computed and the desired Bernoulli distributions. Small positive values of $\alpha$ ensure stable computation of the KL divergence. Recalling that $p_r$ is a function of the model and its parameters, we can now use the modified loss function in place of the original loss in the previously described algorithms. We use the hat symbol ( ˆ ) to denote unlearning algorithms that operate with the additional loss term (9). For example, gradient surgery (S), UNO, and UNO-S become Ŝ, UNÔ, and UNÔ-Ŝ, respectively, when (9) is utilized. Addition of the new term yields the following modified definitions of $\mathbf{g_f}$ and $\mathbf{g_r}$:

$$\mathbf{g_f} = \frac{1}{|\mathcal{D}_f|} \sum_{x \in \mathcal{D}_f} \nabla_\theta \mathcal{L}(\mathcal{M}_\theta, x) + \beta_h \nabla_\theta d_{\mathrm{KL}}, \tag{10}$$

$$\mathbf{g_r} = \frac{1}{|\mathcal{D}_r|} \sum_{x \in \mathcal{D}_r} \nabla_\theta \mathcal{L}(\mathcal{M}_\theta, x) + \beta_h \nabla_\theta d_{\mathrm{KL}}. \tag{11}$$

Using (10), (11) with (S) gives us Ŝ. Similarly, the update rule for UNÔ can be written as,

$$\mathcal{L}_{\text{UNÔ}} = \frac{1}{|\mathcal{D}_r|} \sum_{x \in \mathcal{D}_r} \mathcal{L}(\mathcal{M}_\theta, x) + \beta_o \left( \frac{\mathbf{g_r} \cdot \mathbf{g_f}}{\|\mathbf{g_r}\|\|\mathbf{g_f}\|} \right)^2 + \beta_h d_{\text{KL}}, \qquad \text{(UNÔ)}$$

$$\theta_{k+1} = \theta_k - \eta \nabla_{\theta_k} \mathcal{L}_{\text{UNÔ}}.$$

Alternating update steps of UNÔ and Ŝ gives us UNÔ-Ŝ. Since the KL divergence term promotes unlearning of the forget data by preventing generation of forget samples, we also test the following update rule which is equivalent to UNÔ with $\beta_o = 0$,

$$\mathcal{L}_H = \frac{1}{|\mathcal{D}_r|} \sum_{x \in \mathcal{D}_r} \mathcal{L}(\mathcal{M}_\theta, x) + \beta_h d_{\text{KL}}, \qquad \text{(H)}$$

$$\theta_{k+1} = \theta_k - \eta \nabla_{\theta_k} \mathcal{L}_H.$$

We call the resulting unlearning algorithm histogram unlearning and denote it by H. Appendix H reports results for classifier-assisted unlearning on MNIST and CelebA.

## 4 RESULTS

We test the algorithms described in Section 3 and Appendix B on VAEs trained on MNIST (Deng, 2012) and CelebA (Liu et al., 2015), and on diffusion transformers trained on ImageNet-1K (Deng et al., 2009). Each algorithm was tested 10 times to generate statistics. For the training losses used to train the original VAEs, training data, experiment hyperparameters, and model sizes, refer to Appendix A. The architecture of the models can be found in the code provided in Section 1. All experiments were done on an A100 GPU provided by Google Colab.

### 4.1 PERFORMANCE METRICS

In order to assess the speed of unlearning we use classifiers trained on the datasets and track the fraction of generated samples that are classified as forget samples after each model update or training step. We define the **time to unlearn** as the execution time of the unlearning algorithm until the fraction of forget samples in the generated data drops below a chosen threshold $\tau$. On MNIST and CelebA, our classifiers achieve $\sim 98\%$ top-1 accuracy, whereas on ImageNet-1K our classifier achieves $\sim 82\%$ top-1 accuracy. Accordingly, we set $\tau = 0.02$ for MNIST and CelebA and $\tau = 0.18$ for ImageNet-1K. Note that we only consider the execution time of loss computation, gradient calculation, and parameter updates, while excluding auxiliary operations such as data loading and preprocessing. We evaluate the quality of the generated images by computing the **Fréchet Inception Distance (FID)**. We also report the execution **time per training step**, however, we do not highlight these values in the tables, as a larger time per step does not necessarily indicate slower unlearning, and vice versa.

### 4.2 MNIST

We use a 0.6M-parameter VAE with a 2-dimensional latent space, trained for 200 epochs on $60,000$ images, as our original model. We attempt to unlearn the digit "1" by running the algorithms for 530 training steps with a mini-batch size of 128, and a learning rate of $10^{-3}$. Figure 1 shows samples generated before and after unlearning with UNO, using the same noise samples for ease of comparison. The 1's in the original generation (left) transform into $7, 8$ and $3$ after unlearning (right). The non-1 digits remain nearly unchanged. Even though 1's can transform into many different digits, they have an affinity for turning into 8's, followed by 3's, as seen in Figure 2. This can be explained by examining the distribution and proximity of the digits in the latent space, see Appendix E for a detailed discussion. If the goal is uniform generation across the retain classes, one may utilize a Kullback-Leibler divergence loss term promoting uniformity, assuming the availability of a classifier for all classes.

Table 1 shows that UNO-S achieves the fastest unlearning time, closely followed by UNO, with both having similar fidelity as the original model, indicated by the FID. Gradient ascent, while fast at unlearning, suffers from catastrophic forgetting, resulting in a large FID. Ascent-descent also

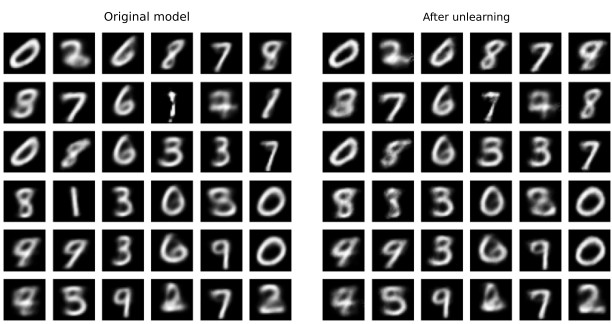

Figure 1: MNIST samples generated by the original model (left) and after unlearning digit "1" with UNO (right), using identical noise inputs for the decoder.

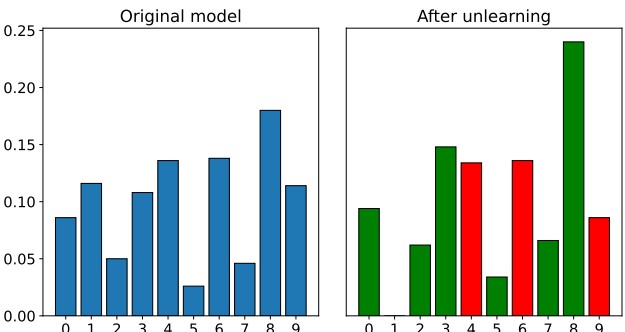

Figure 2: Distribution of generated digits before (left) and after unlearning (right), for a single run of UNO. Each histogram shows data for 500 generated samples. A bar in the right panel is colored green if the fraction of the corresponding digit increases after unlearning, and red if it decreases.

experiences catastrophic forgetting and is significantly slower at unlearning than gradient ascent. Gradient surgery, while preserving image quality, is $\sim 20$ times slower than UNO and UNO-S at unlearning. Even though UNO takes $\sim 3$ times longer to execute a training step compared to gradient surgery, it still achieves orders of magnitude faster unlearning speed. Since one step of surgery is faster than one step of UNO, UNO-S overall is slightly faster than UNO, as the time per training step is roughly averaged over the two algorithms.

## 4.3 CELEBA

We use an $8.7$M-parameter VAE with a $512$-dimensional latent space, trained for 200 epochs on $202,599$ images at $64 \times 64$ resolution, downsampled from the original $178 \times 178$ resolution, as our original model. We attempt to unlearn "male" faces by running the algorithms for $659$ training steps with a mini-batch size of $128$, and a learning rate of $10^{-3}$. Approximately $29\%$ of the faces generated by the original model are male. Figure 3 shows samples generated before and after unlearning with UNO, using the same noise samples in the decoder. We observe that male faces are successfully converted into female faces, and that feminine features are enhanced after unlearning, even when the originally generated face was already female. The original image remains nearly unchanged if it contains few or no male-specific features; see, for example, the last pair from the left in Figure 3. One notable effect of unlearning male-specific features is that the transformed images exhibit broader smiles. This is due to the sociological phenomenon wherein women tend to smile more than men in photographs (Wondergem & Friedlmeier, 2012). Furthermore, we detect an increase of eye make-up in images of females after unlearning. For more examples of these effects, see a larger collection of before/after unlearning pairs in Appendix F, where we also show an example of unlearning eyeglasses.

Table 1 shows that UNO-S again achieves the fastest time to unlearn, followed by UNO. Even after spending $\sim 20$ times more execution time than the time to unlearn with UNO, gradient surgery

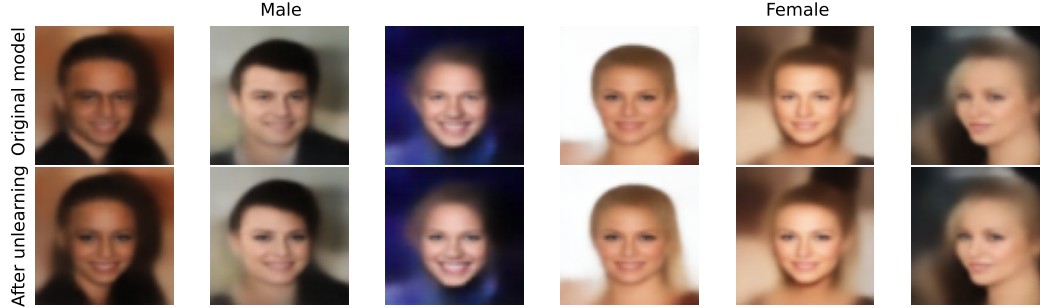

Figure 3: CelebA samples generated by the original model (top) and after unlearning "male" faces with UNO (bottom), using identical noise inputs for the decoder.

is unable to achieve the desired $\leq 2\%$ male faces in the generated images. After the $659$ allotted training steps gradient surgery is only able to reach $\sim 4\%$ male faces. All three algorithms result in similar values of FID, and the quality of the generated images is perceptually indistinguishable from the originally generated images, as seen in Figure 3.

### 4.4 IMAGENET-1K

We use a 675M-parameter diffusion transformer DiT-XL/2 (Peebles & Xie, 2023) that operates on a $4 \times 32 \times 32$-dimensional latent space, trained for 7M steps on $1.28$M images at $256 \times 256$ resolution. We attempt to unlearn class 207 (Golden Retriever) by the algorithms in Section 3.4 for 100 training steps with a mini-batch size of $10$, and a learning rate of $10^{-4}$. We map the class 207 (Golden Retriever) to images of labrador retrievers (class 208), i.e. $c_t = 208$ and $c_f = 207$. This reduces the training to only consider two classes, rather than the full data set. To compute the time to unlearn, at each training step we generate samples only for the class 207 with a (classifier-free) guidance scale of $8$ and determine what fraction of the samples are classified as golden retrievers.

Figure 4 shows that noise in the latent space that generated golden retrievers in the original model, generate labrador retrievers after unlearning with conditional UNO-S. Images belonging to other classes are significantly more changed than for CelebA, but remain in their respective classes. A larger collection of before/after unlearning pairs are provided in Appendix G.

Our algorithms outperform gradient surgery both on time to unlearn and FID (see Table 1). For ImageNet-1K the differences in performance are less pronounced. We believe that this is due to the simpler set-up of conditional unlearning with only two classes. We remark that Peebles & Xie (2023) report an FID value of approximately $2.3$ whereas our FID values are around $12$. This is due to their significantly larger sample size of $50,000$ compared to our $22,000$ samples.

## 5 DISCUSSION

We advance the gradient surgery paradigm for machine unlearning by introducing two new algorithms UNO and UNO-S and their conditional variants. We show that they are as fast as gradient ascent at unlearning but without suffering from catastrophic forgetting, and are substantially faster than gradient surgery for unconditional generative models. UNO-S outperforms all other algorithms for unconditional generative models, and can be up to $1.3$ times faster than UNO at unlearning. For conditional unlearning UNO marginally outperformed UNO-S. We have shown the efficiency of our algorithms for data sets and generative models of increasing complexity. We demonstrate how incorporating the information provided by a classifier that distinguishes between desirable and undesired data, can accelerate unlearning algorithms (see Appendix H). Table 2 in Appendix A documents the hyperparameters used in our experiments. Our experiments indicate that UNO and UNO-S are robust with respect to the selection of hyperparameters; in particular, for MNIST and CelebA we use identical hyperparameter values (cf. Table 2).

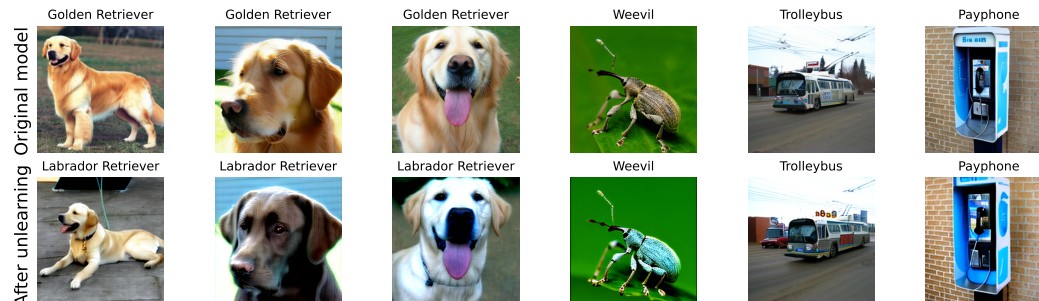

Figure 4: ImageNet samples generated by the original model (top) and after unlearning class 207 (Golden Retriever) with UNO-S (bottom), using identical noise inputs for the diffusion transformer. The labels classifying each image was provided by the pretrained classifier.

Table 1: Performance of various algorithms for class/feature unlearning with VAEs on MNIST and CelebA and diffusion transformers on ImageNet-1K. Each experiment is repeated $10$ times, and the standard deviations are shown in parentheses. FID values are calculated using $25,000$ samples for MNIST and CelebA and $22,000$ samples for ImageNet-1K. ✗ indicates that the generated samples after unlearning are unrecognizably different from the original model. Bold indicates the best score among instances with acceptable FID. ✓ indicates the generated samples after unlearning are perceptually indistinguishable from the original model in terms of visual fidelity. An asterisk (*) denotes cases where the algorithm failed to reach the target fraction of forget samples in the generated images within the allotted training steps. We only report on (A) and (A-D) for MNIST since they demonstrate catastrophic forgetting in all cases; for CelebA (A) and (A-D) result in NaN for FID, generating white images.

| Dataset | Algorithm | Time to unlearn (s) ↓ | FID ↓ | Time per step (s) |
|---|---|---|---|---|
| MNIST (Class: 1) Original FID: 20.7 | Gradient ascent (A) | 0.024 (0.004) | 612.3 (4.9) ✗ | 0.005 (0.0005) |
| | Ascent descent (A-D) | 0.025 (0.006) | 266.9 (19.3) ✗ | 0.005 (0.0001) |
| | Gradient surgery (S) | 1.016 (0.907) | 23.0 (0.2) ✓ | 0.007 (0.0001) |
| | UNO | 0.055 (0.009) | **21.8** (0.2) ✓ | 0.020 (0.0003) |
| | UNO-S | **0.041** (0.010) | **21.8** (0.4) ✓ | 0.015 (0.0003) |
| CelebA (Feature: Male) Original FID: 166.3 | Gradient surgery (S) | 10.71* (3.36) | 176.0 (3.7) ✓ | 0.018 (0.0005) |
| | UNO | 0.524 (0.002) | **174.3** (1.6) ✓ | 0.175 (0.0007) |
| | UNO-S | **0.414** (0.186) | 177.1 (4.7) ✓ | 0.148 (0.0625) |
| ImageNet-1K (Class: 207) Original FID: 12.0 | Gradient surgery (S) | 7.622 (2.21) | 12.3 (0.8) ✓ | 0.712 (0.0090) |
| | UNO | **6.361** (0.909) | **11.9** (0.8) ✓ | 1.817 (0.0046) |
| | UNO-S | 6.495 (1.928) | 12.0 (0.8) ✓ | 1.273 (0.0080) |

## FUTURE WORK

It is straightforward to conceptualize low-rank adapted (Hu et al., 2021; Xu et al., 2023) variants of the unlearning algorithms presented here. Such modifications are essential for enabling efficient unlearning in large-scale generative models, and we leave their exploration to future research. The CelebA experiments show that, unlearning can easily produce male-to-female face filters. Applications of unlearning for designing a broader range of filters is an interesting topic for further exploration. Machine learning models used to simulate or predict physical systems, such as climate models, often generate unphysical states (Lai et al., 2024). A similar issue arises in video generation models like Sora (Kang et al., 2024), which can produce physically implausible outputs. Use of unlearning to prevent generation of such unphysical outputs could be explored in future.

## 6 REPRODUCIBILITY STATEMENT

We provide all details necessary to reproduce our work. The paper describes the algorithms, training procedure, hyperparameters, and datasets in detail. Code to reproduce the experiments has been provided via an anonymized file. Experiments were conducted on a A100 GPU with 80GB GPU RAM. We also report standard deviations over multiple runs for our results to ensure robustness.

## 7 ETHICS STATEMENT

We abide by the ICLR Code of Ethics and are not aware of any direct ethical concerns associated with this work.

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

## A   EXPERIMENT SETUP

In this section we provide additional details of the experimental setup used to produce the reported results.

### A.1   VAE LOSS FUNCTIONS AND TRAINING DATA

We document here the loss functions used to train the original model. For each input image $x$, the encoder outputs $\mu(x) \in \mathbb{R}^{d_z}$ and $\sigma(x) \in \mathbb{R}^{d_z}$, which parameterize the approximate posterior distribution. Here $d_z$ is the latent dimension. The corresponding reconstruction of $x$ by the decoder is denoted by $\bar{x}$, with $\bar{x}_i$ referring to its $i$-th pixel.

The VAE used as the original model for MNIST was trained using the loss function

$$\mathcal{L}_{\text{MNIST}} = \frac{1}{|\mathcal{D}|} \sum_{x \in \mathcal{D}} \Big[ - \sum_{i=1}^{784} (x_i \log(\bar{x}_i) + (1 - x_i) \log(1 - \bar{x}_i)) \\ + \frac{1}{2} \sum_{i=1}^{d_z} \left( \mu_i^2(x) + \sigma_i^2(x) - \log \sigma_i^2(x) - 1 \right) \Big]. \tag{12}$$

The $60,000$ training images were normalized such that pixel values lie in $[0, 1]$, following standard practice.

The VAE used as the original model for CelebA was trained using the loss function

$$\mathcal{L}_{\text{CelebA}} = \frac{1}{|\mathcal{D}|} \sum_{x \in \mathcal{D}} \left[ \|x - \bar{x}\|^2 + \frac{1}{2} \sum_{i=1}^{d_z} \left( \mu_i^2(x) + \sigma_i^2(x) - \log \sigma_i^2(x) - 1 \right) \right]. \tag{13}$$

We worked with $202,599$ cropped and aligned images in CelebA which originally have resolution $178 \times 178$ pixels. We downsampled these images to $64 \times 64$ resolution for training.

### A.2   HYPERPARAMETERS

Table 2 lists the hyperparameters used in the unlearning experiments presented here. Here $\eta$ is the learning rate, $K$ is the number of training steps executed, $\beta_o$ is the weight for the orthogonalization loss term in (3), $\beta_h$ is the weight for the KL divergence loss term in (9), $\alpha$ is a small positive threshold for stable computation of the KL divergence in (9), $B$ is the batch size, and $N_{\text{FID}}$ is the number of samples used for calculating FID. We use $N_g = B$ for all the algorithms in Appendix B.4, which determines the number of samples to be generated using the generative model. Each method was tested 10 times for each dataset. For MNIST, FID was computed using features extracted from the classifier model, whereas for CelebA, features were computed using the InceptionV3 model (Szegedy et al., 2016). All experiments were done on an A100 GPU provided by Google Colab.

### A.3   MODEL SIZES

The VAE models for MNIST and CelebA have 632,788 and 8,742,659 parameters with latent dimension $d_z = 2$ and $d_z = 512$, respectively. The classifier models for MNIST and CelebA have 159,410 and 2,190,913 parameters, respectively. For the exact model implementations, please refer to the code linked in Section 1. The VAEs were trained for 200 epochs on $60,000$ and $202,599$ images in MNIST and CelebA, respectively. The classifiers were trained for 10 epochs on these datasets. For ImageNet-1K, we use the DiT-XL/2 diffusion transformer with 675.13M parameters, trained for 7M steps on $256 \times 256$ images; see (Peebles & Xie, 2023) for details. For the ImageNet-1K classifier, we use microsoft/swinv2-tiny-patch4-window8-256 (Liu et al., 2022), which has 28M parameters.

Table 2: Experiment hyperparameters

| Dataset | Algorithm | $\eta$ | $K$ | $B\beta_o$ | $B\beta_h$ | $\alpha$ | $B$ | $N_{\text{FID}}$ |
|---|---|---|---|---|---|---|---|---|
| MNIST (Class: 1) | Gradient ascent (A) | $10^{-3}$ | 530 | - | - | - | 128 | 25,000 |
| | Ascent descent (A-D) | $10^{-3}$ | 530 | - | - | - | 128 | 25,000 |
| | Gradient surgery (S) | $10^{-3}$ | 530 | - | - | - | 128 | 25,000 |
| | UNO | $10^{-3}$ | 530 | $10^3$ | - | - | 128 | 25,000 |
| | UNO-S | $10^{-3}$ | 530 | $10^3$ | - | - | 128 | 25,000 |
| | H | $10^{-3}$ | 530 | - | $10^3$ | $10^{-8}$ | 128 | 25,000 |
| | Ŝ | $10^{-3}$ | 530 | - | $10^3$ | $10^{-8}$ | 128 | 25,000 |
| | UNÔ | $10^{-3}$ | 530 | $10^3$ | $10^3$ | $10^{-8}$ | 128 | 25,000 |
| | UNÔ-Ŝ | $10^{-3}$ | 530 | $10^3$ | $10^3$ | $10^{-8}$ | 128 | 25,000 |
| CelebA (Feature: Male) | Gradient surgery (S) | $10^{-3}$ | 659 | - | - | - | 128 | 25,000 |
| | UNO | $10^{-3}$ | 659 | $10^3$ | - | - | 128 | 25,000 |
| | UNO-S | $10^{-3}$ | 659 | $10^3$ | - | - | 128 | 25,000 |
| | H | $10^{-3}$ | 659 | - | $10^3$ | $10^{-8}$ | 128 | 25,000 |
| | Ŝ | $10^{-3}$ | 659 | - | $10^3$ | $10^{-8}$ | 128 | 25,000 |
| | UNÔ | $10^{-3}$ | 659 | $10^3$ | $10^3$ | $10^{-8}$ | 128 | 25,000 |
| | UNÔ-Ŝ | $10^{-3}$ | 659 | $10^3$ | $10^3$ | $10^{-8}$ | 128 | 25,000 |
| ImageNet-1K (Class: 207) | Gradient surgery (S) | $10^{-4}$ | 100 | - | - | - | 10 | 22,000 |
| | UNO | $10^{-4}$ | 100 | $2 \times 10^{-2}$ | - | - | 10 | 22,000 |
| | UNO-S | $10^{-4}$ | 100 | $2 \times 10^{-2}$ | - | - | 10 | 22,000 |

## B PSEUDOCODE FOR UNLEARNING ALGORITHMS

This section presents the pseudocode for the unlearning algorithms used in this work.

### B.1 GRADIENT ASCENT

Algorithms 1 and 2 describe the gradient ascent (A), and alternating gradient ascent-descent (A-D), respectively.

### B.2 GRADIENT SURGERY

Algorithms 3 and 4 describe gradient surgery with ascent in the forget direction (SA) and descent in the retain direction (S), respectively; in particular, the former appears in Bae et al. (2023).

### B.3 UNO AND UNO-S

Algorithms 5 and 6 describe unlearning via orthogonalization (UNO), and alternating orthogonalization and surgery (UNO-S), respectively.

### B.4 CLASSIFIER-ASSISTED UNLEARNING

Algorithms 7, 8, 9, and 10 describe Ŝ, UNÔ, UNÔ-Ŝ, and histogram unlearning, respectively. While in practice it is sufficient for a binary classifier to output a logit or probability, for simplicity of presentation we assume the classifier outputs 1 for retain samples and 0 otherwise.

**Algorithm 1** Gradient ascent (A)

1: **Input:** Loss function $\mathcal{L}$, forget dataset $\mathcal{D}_f$, trained model requiring unlearning $\mathcal{M}_\theta$, learning rate $\eta$, number of training steps $K$, batch size $B$.
2: **Output:** Updated model parameters $\theta$.
3: **for** $k = 1$ to $K$ **do**
4:     Acquire mini-batch $D_f$ of size $B$ from $\mathcal{D}_f$.
5:     $\mathbf{g_f} \leftarrow \frac{1}{B} \sum_{x \in D_f} \nabla_\theta \mathcal{L}(\mathcal{M}_\theta, x)$
6:     $\theta \leftarrow \theta + \eta \mathbf{g_f}$
7: **end for**
8: **return** $\theta$

---

**Algorithm 2** Alternating gradient ascent and descent (A-D)

1: **Input:** Loss function $\mathcal{L}$, retain dataset $\mathcal{D}_r$, forget dataset $\mathcal{D}_f$, trained model requiring unlearning $\mathcal{M}_\theta$, learning rate $\eta$, number of training steps $K$, batch size $B$.
2: **Output:** Updated model parameters $\theta$.
3: **for** $k = 1$ to $K$ **do**
4:     Acquire retain and forget mini-batches $D_r, D_f$ of size $B$ from $\mathcal{D}_r, \mathcal{D}_f$ respectively.
5:     **if** $k$ is odd **then**
6:         $\mathbf{g_f} \leftarrow \frac{1}{B} \sum_{x \in D_f} \nabla_\theta \mathcal{L}(\mathcal{M}_\theta, x)$
7:         $\theta \leftarrow \theta + \eta \mathbf{g_f}$
8:     **else**
9:         $\mathbf{g_r} \leftarrow \frac{1}{B} \sum_{x \in D_r} \nabla_\theta \mathcal{L}(\mathcal{M}_\theta, x)$
10:        $\theta \leftarrow \theta - \eta \mathbf{g_r}$
11:     **end if**
12: **end for**
13: **return** $\theta$

---

**Algorithm 3** Gradient surgery with ascent in forget direction (SA)

1: **Input:** Loss function $\mathcal{L}$, retain dataset $\mathcal{D}_r$, forget dataset $\mathcal{D}_f$, trained model requiring unlearning $\mathcal{M}_\theta$, learning rate $\eta$, number of training steps $K$, batch size $B$.
2: **Output:** Updated model parameters $\theta$.
3: **for** $k = 1$ to $K$ **do**
4:     Acquire retain and forget mini-batches $D_r, D_f$ of size $B$ from $\mathcal{D}_r, \mathcal{D}_f$ respectively.
5:     $\mathbf{g_r} \leftarrow \frac{1}{B} \sum_{x \in D_r} \nabla_\theta \mathcal{L}(\mathcal{M}_\theta, x)$
6:     $\mathbf{g_f} \leftarrow \frac{1}{B} \sum_{x \in D_f} \nabla_\theta \mathcal{L}(\mathcal{M}_\theta, x)$
7:     $\mathbf{g_f} \leftarrow \mathbf{g_f} - \frac{\mathbf{g_r} \cdot \mathbf{g_f}}{\|\mathbf{g_r}\|^2} \mathbf{g_r}$
8:     $\theta \leftarrow \theta + \eta \mathbf{g_f}$
9: **end for**
10: **return** $\theta$

---

**Algorithm 4** Gradient surgery with descent in retain direction (S)

1: **Input:** Loss function $\mathcal{L}$, retain dataset $\mathcal{D}_r$, forget dataset $\mathcal{D}_f$, trained model requiring unlearning $\mathcal{M}_\theta$, learning rate $\eta$, number of training steps $K$, batch size $B$.
2: **Output:** Updated model parameters $\theta$.
3: **for** $k = 1$ to $K$ **do**
4:     Acquire retain and forget mini-batches $D_r, D_f$ of size $B$ from $\mathcal{D}_r, \mathcal{D}_f$ respectively.
5:     $\mathbf{g_r} \leftarrow \frac{1}{B} \sum_{x \in D_r} \nabla_\theta \mathcal{L}(\mathcal{M}_\theta, x)$
6:     $\mathbf{g_f} \leftarrow \frac{1}{B} \sum_{x \in D_f} \nabla_\theta \mathcal{L}(\mathcal{M}_\theta, x)$
7:     $\mathbf{g_r} \leftarrow \mathbf{g_r} - \frac{\mathbf{g_r} \cdot \mathbf{g_f}}{\|\mathbf{g_f}\|^2} \mathbf{g_f}$
8:     $\theta \leftarrow \theta - \eta \mathbf{g_r}$
9: **end for**
10: **return** $\theta$

**Algorithm 5** Unlearning via orthogonalization (UNO)

1: **Input:** Loss function $\mathcal{L}$, retain dataset $\mathcal{D}_r$, forget dataset $\mathcal{D}_f$, trained model requiring unlearning $\mathcal{M}_\theta$, weight for orthogonalization loss term $\beta_o$, learning rate $\eta$, number of training steps $K$, batch size $B$.
2: **Output:** Updated model parameters $\theta$.
3: **for** $k = 1$ to $K$ **do**
4:     Acquire retain and forget mini-batches $D_r, D_f$ of size $B$ from $\mathcal{D}_r, \mathcal{D}_f$ respectively.
5:     $L_r \leftarrow \frac{1}{B} \sum_{x \in D_r} \mathcal{L}(\mathcal{M}_\theta, x)$
6:     $\mathbf{g_r} \leftarrow \nabla_\theta L_r$
7:     $\mathbf{g_f} \leftarrow \frac{1}{B} \sum_{x \in D_f} \nabla_\theta \mathcal{L}(\mathcal{M}_\theta, x)$
8:     $L \leftarrow L_r + \beta_o \left( \frac{\mathbf{g_r} \cdot \mathbf{g_f}}{\|\mathbf{g_r}\| \|\mathbf{g_f}\|} \right)^2$
9:     $\theta \leftarrow \theta - \eta \nabla_\theta L$
10: **end for**
11: **return** $\theta$

---

**Algorithm 6** Alternating orthogonalization and surgery (UNO-S)

1: **Input:** Loss function $\mathcal{L}$, retain dataset $\mathcal{D}_r$, forget dataset $\mathcal{D}_f$, trained model requiring unlearning $\mathcal{M}_\theta$, weight for orthogonalization loss term $\beta_o$, learning rate $\eta$, number of training steps $K$, batch size $B$.
2: **Output:** Updated model parameters $\theta$.
3: **for** $k = 1$ to $K$ **do**
4:     Acquire retain and forget mini-batches $D_r, D_f$ of size $B$ from $\mathcal{D}_r, \mathcal{D}_f$ respectively.
5:     $L_r \leftarrow \frac{1}{B} \sum_{x \in D_r} \mathcal{L}(\mathcal{M}_\theta, x)$
6:     $\mathbf{g_r} \leftarrow \nabla_\theta L_r$
7:     $\mathbf{g_f} \leftarrow \frac{1}{B} \sum_{x \in D_f} \nabla_\theta \mathcal{L}(\mathcal{M}_\theta, x)$
8:     **if** $k$ is odd **then**
9:         $L \leftarrow L_r + \beta_o \left( \frac{\mathbf{g_r} \cdot \mathbf{g_f}}{\|\mathbf{g_r}\| \|\mathbf{g_f}\|} \right)^2$
10:         $\theta \leftarrow \theta - \eta \nabla_\theta L$
11:     **else**
12:         $\mathbf{g_r} \leftarrow \mathbf{g_r} - \frac{\mathbf{g_r} \cdot \mathbf{g_f}}{\|\mathbf{g_f}\|^2} \mathbf{g_f}$
13:         $\theta \leftarrow \theta - \eta \mathbf{g_r}$
14:     **end if**
15: **end for**
16: **return** $\theta$

---

**Algorithm 7** Gradient surgery with histogram unlearning ($\hat{\mathrm{S}}$)

1: **Input:** Loss function $\mathcal{L}$, retain dataset $\mathcal{D}_r$, forget dataset $\mathcal{D}_f$, trained model requiring unlearning $\mathcal{M}_\theta$, learning rate $\eta$, number of training steps $K$, batch size $B$, number of samples to generate $N_g$, classifier model $\mathcal{C}_\phi$, weight for KL divergence loss term $\beta_h$, a small positive threshold for stabilizing KL divergence computation $\alpha$.
2: **Output:** Updated model parameters $\theta$.
3: **for** $k = 1$ to $K$ **do**
4:     Acquire retain and forget mini-batches $D_r, D_f$ of size $B$ from $\mathcal{D}_r, \mathcal{D}_f$ respectively.
5:     Generate $N_g$ samples $\{y_i\}_{i=1}^{N_g}$ using $\mathcal{M}_\theta$.
6:     $p_r \leftarrow \frac{1}{N_g} \sum_{i=1}^{N_g} \mathcal{C}_\phi(y_i)$
7:     $d_{\mathrm{KL}} \leftarrow p_r \log \left( \frac{p_r}{1-\alpha} \right) + (1 - p_r) \log \left( \frac{1-p_r}{\alpha} \right)$
8:     $\mathbf{g_r} \leftarrow \frac{1}{B} \sum_{x \in D_r} \nabla_\theta \mathcal{L}(\mathcal{M}_\theta, x) + \beta_h \nabla_\theta d_{\mathrm{KL}}$
9:     $\mathbf{g_f} \leftarrow \frac{1}{B} \sum_{x \in D_f} \nabla_\theta \mathcal{L}(\mathcal{M}_\theta, x) + \beta_h \nabla_\theta d_{\mathrm{KL}}$
10:     $\mathbf{g_r} \leftarrow \mathbf{g_r} - \frac{\mathbf{g_r} \cdot \mathbf{g_f}}{\|\mathbf{g_f}\|^2} \mathbf{g_f}$
11:     $\theta \leftarrow \theta - \eta \mathbf{g_r}$
12: **end for**
13: **return** $\theta$

**Algorithm 8** UNO with histogram unlearning (UNÔ)

---

1: **Input:** Loss function $\mathcal{L}$, retain dataset $\mathcal{D}_r$, forget dataset $\mathcal{D}_f$, trained model requiring unlearning $\mathcal{M}_\theta$, weight for orthogonalization loss term $\beta_o$, learning rate $\eta$, number of training steps $K$, batch size $B$, number of samples to generate $N_g$, classifier model $\mathcal{C}_\phi$, weight for KL divergence loss term $\beta_h$, a small positive threshold for stabilizing KL divergence computation $\alpha$.
2: **Output:** Updated model parameters $\theta$.
3: **for** $k = 1$ to $K$ **do**
4:      Acquire retain and forget mini-batches $D_r, D_f$ of size $B$ from $\mathcal{D}_r, \mathcal{D}_f$ respectively.
5:      Generate $N_g$ samples $\{y_i\}_{i=1}^{N_g}$ using $\mathcal{M}_\theta$.
6:      $p_r \leftarrow \frac{1}{N_g} \sum_{i=1}^{N_g} \mathcal{C}_\phi(y_i)$
7:      $d_{\mathrm{KL}} \leftarrow p_r \log\left(\frac{p_r}{1-\alpha}\right) + (1 - p_r)\log\left(\frac{1-p_r}{\alpha}\right)$
8:      $L_r \leftarrow \frac{1}{B} \sum_{x \in D_r} \mathcal{L}(\mathcal{M}_\theta, x) + \beta_h d_{\mathrm{KL}}$
9:      $\mathbf{g_r} \leftarrow \nabla_\theta L_r$
10:     $\mathbf{g_f} \leftarrow \frac{1}{B} \sum_{x \in D_f} \nabla_\theta \mathcal{L}(\mathcal{M}_\theta, x) + \beta_h \nabla_\theta d_{\mathrm{KL}}$
11:     $L \leftarrow L_r + \beta_o \left(\frac{\mathbf{g_r} \cdot \mathbf{g_f}}{\|\mathbf{g_r}\|\|\mathbf{g_f}\|}\right)^2$
12:     $\theta \leftarrow \theta - \eta \nabla_\theta L$
13: **end for**
14: **return** $\theta$

---

**Algorithm 9** Alternating orthogonalization and surgery with histogram unlearning (UNÔ-Ŝ)

---

1: **Input:** Loss function $\mathcal{L}$, retain dataset $\mathcal{D}_r$, forget dataset $\mathcal{D}_f$, trained model requiring unlearning $\mathcal{M}_\theta$, weight for orthogonalization loss term $\beta_o$, learning rate $\eta$, number of training steps $K$, batch size $B$, number of samples to generate $N_g$, classifier model $\mathcal{C}_\phi$, weight for KL divergence loss term $\beta_h$, a small positive threshold for stabilizing KL divergence computation $\alpha$.
2: **Output:** Updated model parameters $\theta$.
3: **for** $k = 1$ to $K$ **do**
4:      Acquire retain and forget mini-batches $D_r, D_f$ of size $B$ from $\mathcal{D}_r, \mathcal{D}_f$ respectively.
5:      Generate $N_g$ samples $\{y_i\}_{i=1}^{N_g}$ using $\mathcal{M}_\theta$.
6:      $p_r \leftarrow \frac{1}{N_g} \sum_{i=1}^{N_g} \mathcal{C}_\phi(y_i)$
7:      $d_{\mathrm{KL}} \leftarrow p_r \log\left(\frac{p_r}{1-\alpha}\right) + (1 - p_r)\log\left(\frac{1-p_r}{\alpha}\right)$
8:      $L_r \leftarrow \frac{1}{B} \sum_{x \in D_r} \mathcal{L}(\mathcal{M}_\theta, x) + \beta_h d_{\mathrm{KL}}$
9:      $\mathbf{g_r} \leftarrow \nabla_\theta L_r$
10:     $\mathbf{g_f} \leftarrow \frac{1}{B} \sum_{x \in D_f} \nabla_\theta \mathcal{L}(\mathcal{M}_\theta, x) + \beta_h \nabla_\theta d_{\mathrm{KL}}$
11:     **if** $k$ is odd **then**
12:        $L \leftarrow L_r + \beta_o \left(\frac{\mathbf{g_r} \cdot \mathbf{g_f}}{\|\mathbf{g_r}\|\|\mathbf{g_f}\|}\right)^2$
13:        $\theta \leftarrow \theta - \eta \nabla_\theta L$
14:     **else**
15:        $\mathbf{g_r} \leftarrow \mathbf{g_r} - \frac{\mathbf{g_r} \cdot \mathbf{g_f}}{\|\mathbf{g_f}\|^2} \mathbf{g_f}$
16:        $\theta \leftarrow \theta - \eta \mathbf{g_r}$
17:     **end if**
18: **end for**
19: **return** $\theta$

---

---

**Algorithm 10** Histogram unlearning (H)

---

1: **Input:** Loss function $\mathcal{L}$, retain dataset $\mathcal{D}_r$, forget dataset $\mathcal{D}_f$, trained model requiring unlearning $\mathcal{M}_\theta$, learning rate $\eta$, number of training steps $K$, batch size $B$, number of samples to generate $N_g$, classifier model $\mathcal{C}_\phi$, weight for KL divergence loss term $\beta_h$, a small positive threshold for stabilizing KL divergence computation $\alpha$.
2: **Output:** Updated model parameters $\theta$.
3: **for** $k = 1$ to $K$ **do**
4: $\quad$ Acquire retain and forget mini-batches $D_r, D_f$ of size $B$ from $\mathcal{D}_r, \mathcal{D}_f$ respectively.
5: $\quad$ Generate $N_g$ samples $\{y_i\}_{i=1}^{N_g}$ using $\mathcal{M}_\theta$.
6: $\quad$ $p_r \leftarrow \frac{1}{N_g} \sum_{i=1}^{N_g} \mathcal{C}_\phi(y_i)$
7: $\quad$ $d_{\text{KL}} \leftarrow p_r \log\left(\frac{p_r}{1-\alpha}\right) + (1 - p_r) \log\left(\frac{1-p_r}{\alpha}\right)$
8: $\quad$ $L \leftarrow \frac{1}{B} \sum_{x \in D_r} \mathcal{L}(\mathcal{M}_\theta, x) + \beta_h d_{\text{KL}}$
9: $\quad$ $\theta \leftarrow \theta - \eta \nabla_\theta L$
10: **end for**
11: **return** $\theta$

---

## C  CATASTROPHIC FORGETTING INDUCED BY GRADIENT ASCENT AND ASCENT−DESCENT

Figure 5 shows the generated samples for unlearning the digit 1 after 49 parameter update steps of gradient ascent (A) and ascent-descent (A-D) at a learning rate of $\eta = 10^{-3}$. In this example, the 1's were successfully forgotten, but all other digits were forgotten as well. In particular, the left panel of Figure 5 shows that the model only remembers the complement of 1's after gradient ascent.

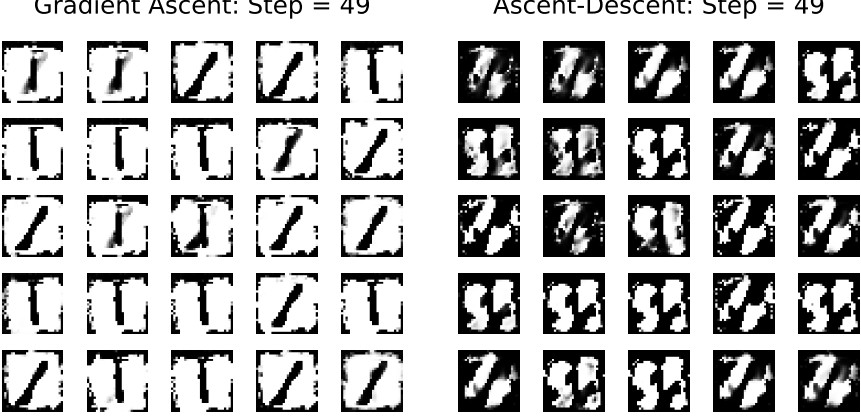

Figure 5: Catastrophic forgetting induced by gradient ascent (left) and ascent descent (right) for MNIST at a learning rate of $10^{-3}$.

## D  COMPARISON OF TWO VARIANTS OF GRADIENT SURGERY

We now compare two variants of gradient surgery: 1) gradient surgery with descent in the retain direction (S), described in Algorithm 4, used throughout this paper and 2) gradient surgery with ascent in the forget direction (SA), described in Algorithm 3 which appears in Bae et al. (2023). Figure 6 shows that SA is prone to catastrophic forgetting and requires a carefully tuned, small learning rate to mitigate this effect. But even with a small learning rate the generated samples might look significantly different from the original model; for samples generated by the original model, see Figure 1. On the other hand, Figure 7 shows that S does not suffer from catastrophic forgetting, even for a large learning rate applied for many training steps, and produces samples that are much closer to the original model.

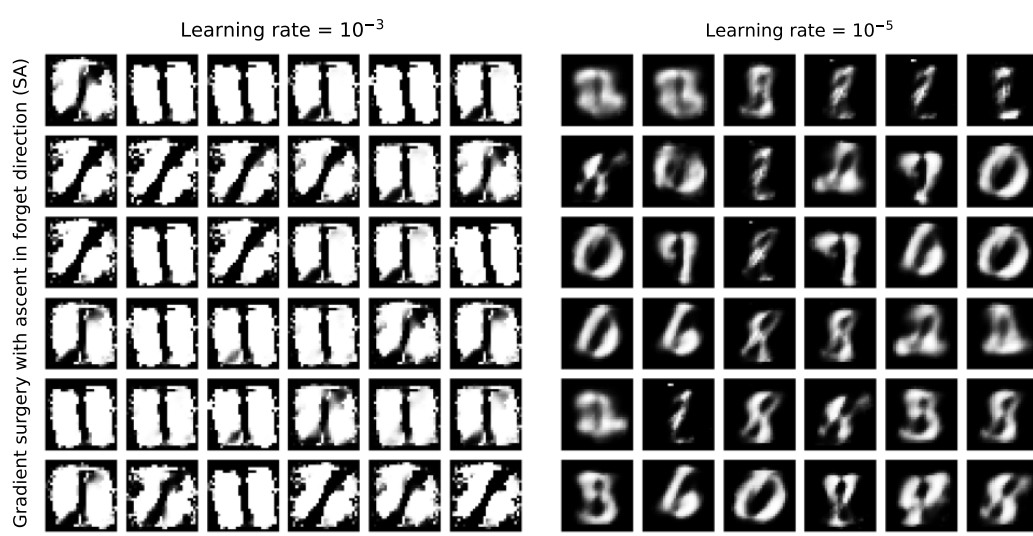

Figure 6: Generated samples after unlearning digit 1 via gradient surgery with ascent in the forget direction (SA), described in Algorithm 3, for two different learning rates: $10^{-3}$ (left), $10^{-5}$ (right). SA was run for $K = 53$ training steps on the left and $K = 530$ training steps on the right.

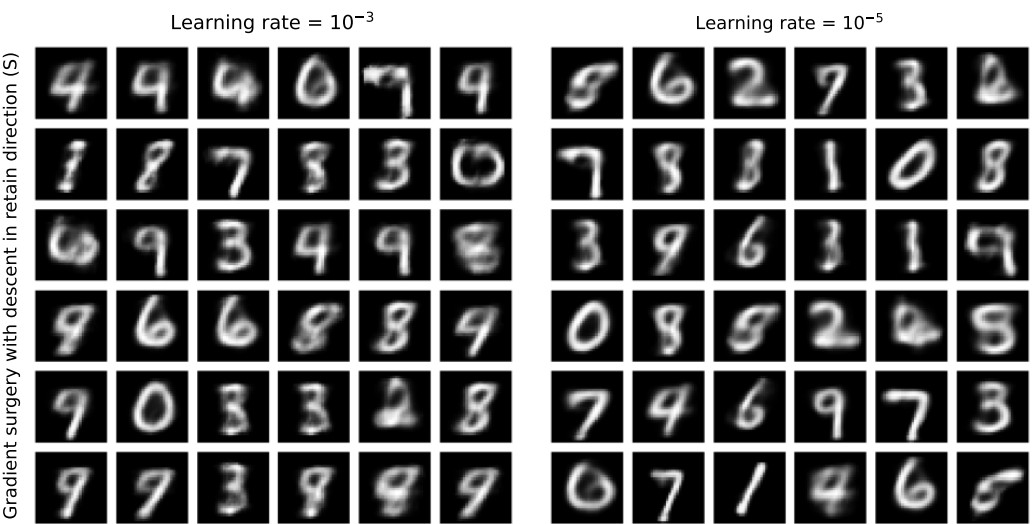

Figure 7: Generated samples after unlearning digit 1 via gradient surgery with descent in the retain direction (S), described in Algorithm 4, for two different learning rates: $10^{-3}$ (left), $10^{-5}$ (right). S was run for $K = 530$ training steps for both learning rates.

## E   LATENT SPACE AND SAMPLE TRANSFORMATION VIA UNLEARNING

In our MNIST example, forgetting the digit 1 leads to an increase of generated digits 8, see Figure 2. To understand this, we color regions in the latent space, which is two-dimensional in our case, according to the most frequently produced digits. Figure 8 shows the distribution of digits in the latent space for the original model. We clearly see that the region corresponding to the digit 1 shares the largest border with the region corresponding to digit 8. This proximity in latent space makes it easier for the unlearning algorithms to transfer the probability mass of 1 to that of 8. Figure 8

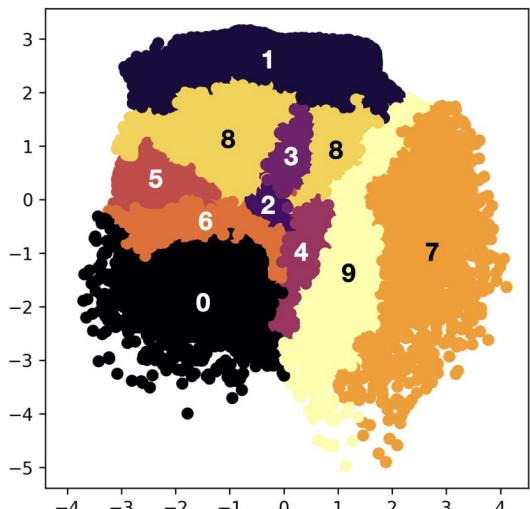

Figure 8: Distribution of digits in the latent space of the original model for MNIST. The colors are obtained by mapping $50,000$ images to the two-dimensional latent space, labelling them according to the color associated with the image. To obtain smooth regions we performed averaging over nearby points in latent space.

suggests that forgetting 7 for our original model should mostly increase the frequency of 9, which we have experimentally verified.

## F ADDITIONAL SAMPLES AND EXPERIMENTS FOR CELEBA

Figure 9 shows 18 pairs of images generated before and after unlearning with UNO for CelebA. In many of these pairs the after image shows a subtly larger smile than the before image, see for example the fourteenth pair. This is consistent with the phenomenon that women tend to smile more than men in photographs (Wondergem & Friedlmeier, 2012).

### F.1 EYEGLASSES REMOVAL WITH UNLEARNING

Out of the $202,599$ images in CelebA, $13,193$ or roughly $6.5\%$ contain faces with eyeglasses. By treating the images with eyeglasses as the forget set and those without eyeglasses as the retain set, we can apply our unlearning algorithms to remove the presence of eyeglasses from the generated samples. Figure 10 shows 18 pairs of generated samples before and after unlearning with UNO-S for the same noise samples used for the decoder. For some of these before–after pairs, where the before image contains opaque, dark eyeglasses, the after image may exhibit darker regions around the eyes, resembling periorbital hyperpigmentation (Sarkar et al., 2016) (see, for example, the last pair, labelled 18). This phenomenon does not occur for images with more transparent eyeglasses. During the original training instance, the model likely conflates the concept of dark eyewear with hyperpigmentation to some extent, due to its finite resolution capabilities. A larger model, capable of learning finer-grained patterns, might better distinguish between these two concepts, and therefore may not produce this phenomenon after unlearning. The hyperparameter values used in these experiments are identical to those reported in Table 2.

## G ADDITIONAL SAMPLES FOR IMAGENET-1K

Figure 11 shows 18 pairs of images generated before and after unlearning with UNO-S for ImageNet-1K. The images exhibit a significant degree of variation, however, they clearly remain in the same class as identified by the pretrained classifier.

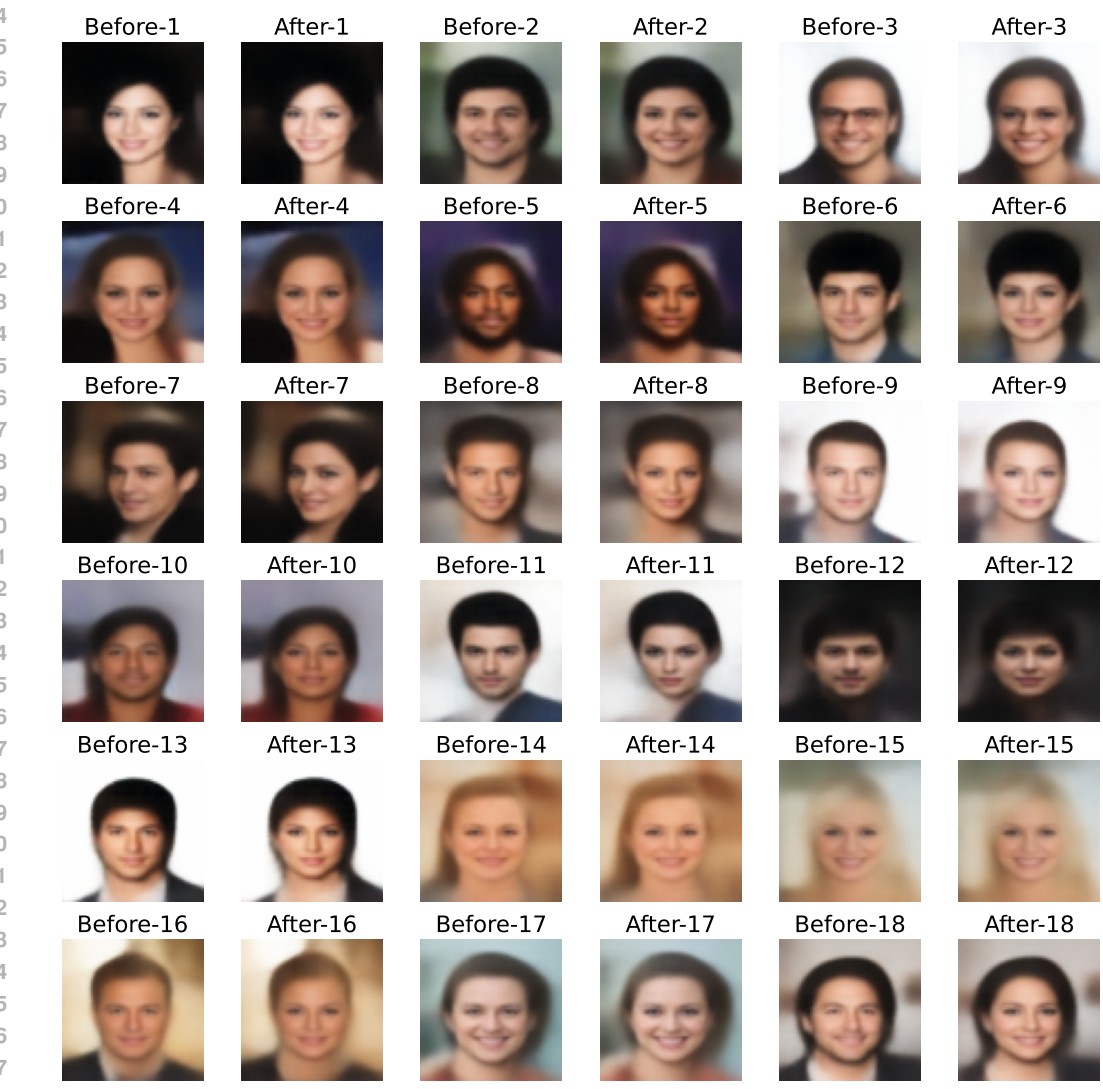

Figure 9: Results for unlearning on CelebA with UNO, illustrated using 18 pairs of generated images. The images labeled "Before" were generated using the original model. Each image labeled "After" was generated after unlearning using the same noise sample for the decoder as the corresponding "Before" image.

## H  RESULTS FOR CLASSIFIER-ASSISTED UNLEARNING

In this section, we present the results for the algorithms introduced in Section 3.5. Comparing Table 1 with Table 3 shows that Ŝ achieves orders of magnitude speed-up over S for both MNIST and CelebA. UNO and UNO-S, already fast, do not gain significant speed-up with classifier-assistance. Histogram unlearning (H), although successful, is much slower than the other successful algorithms in Table 3. All algorithms in Table 3 preserve the fidelity of the original model, with UNÔ producing the lowest FID for both datasets.

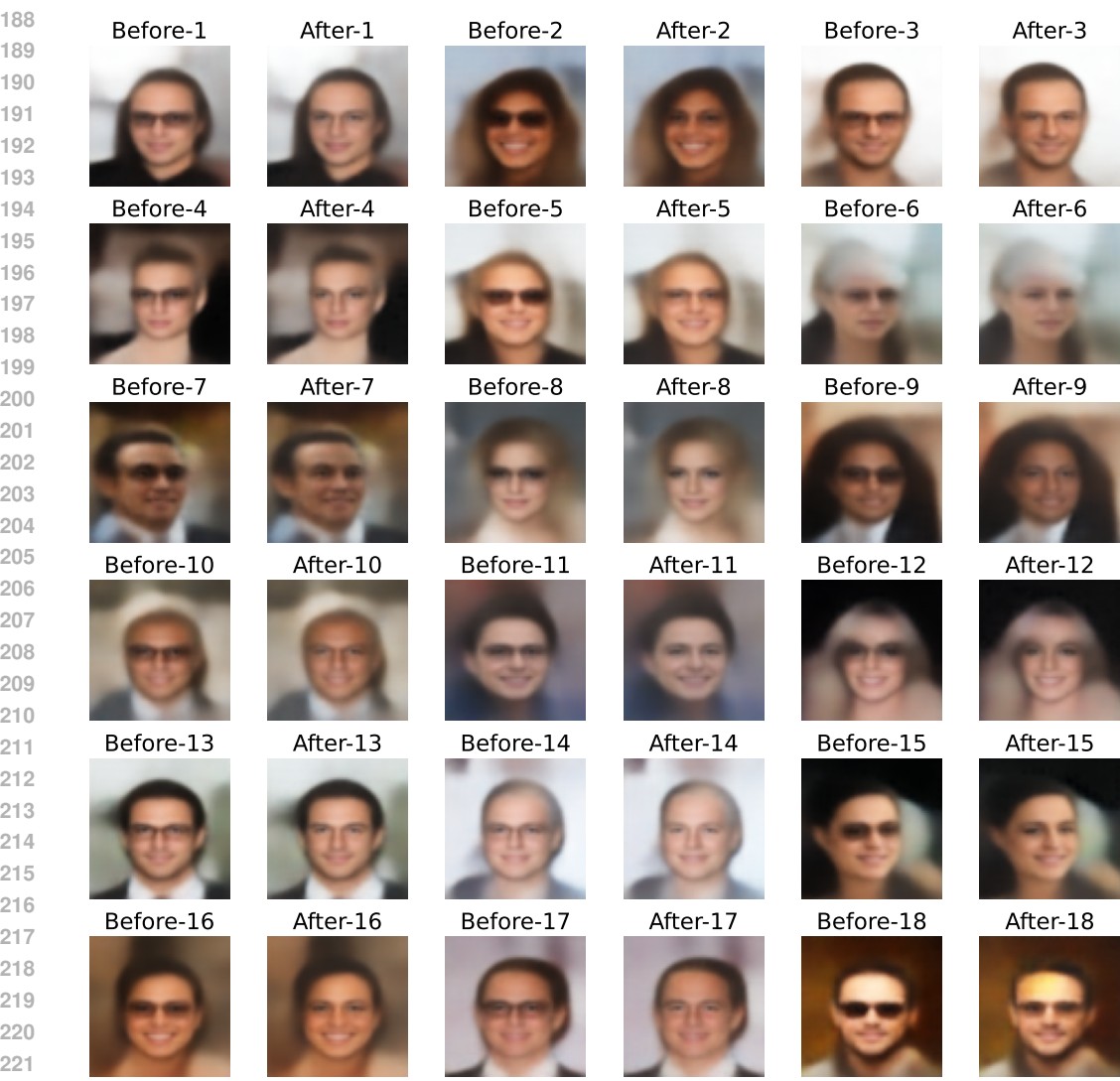

Figure 10: Results for eyeglasses removal on CelebA with UNO-S, illustrated using 18 pairs of generated images. The images labeled "Before" were generated using the original model. Each image labeled "After" was generated after unlearning using the same noise sample for the decoder as the corresponding "Before" image.

# I  ORTHOGONALIZATION IN A LINEAR REGRESSION MODEL

To understand the effect of the orthogonalization term in the loss function (3) let us consider linearly related input $(x)$ and output $(y)$ data

$$y = W^\star x + \zeta, \tag{14}$$

with $\zeta \sim \mathcal{N}(0, \sigma_l^2)$ for $x \in \mathbb{R}^d$, $y \in \mathbb{R}$ and $W \in \mathbb{R}^{1 \times d}$. We consider two data sets, a retain data set and a forget data set, with samples

$$x_r \sim \mathcal{N}(\mu_r, \sigma_r^2) \tag{15}$$

$$x_f \sim \mathcal{N}(\mu_f, \sigma_f^2). \tag{16}$$

Drawing $N_r$ and $N_f$ samples we construct the data matrices $X_r \in \mathbb{R}^{d \times N_r}$ and $X_f \in \mathbb{R}^{d \times N_f}$, and the combined set $X = [X_r \, X_f] \in \mathbb{R}^{d \times N}$ with $N = N_r + N_f$, with corresponding $Y_{r,f} \in \mathbb{R}^{1 \times N_{r,f}}$ and $Y \in \mathbb{R}^{1 \times N}$.

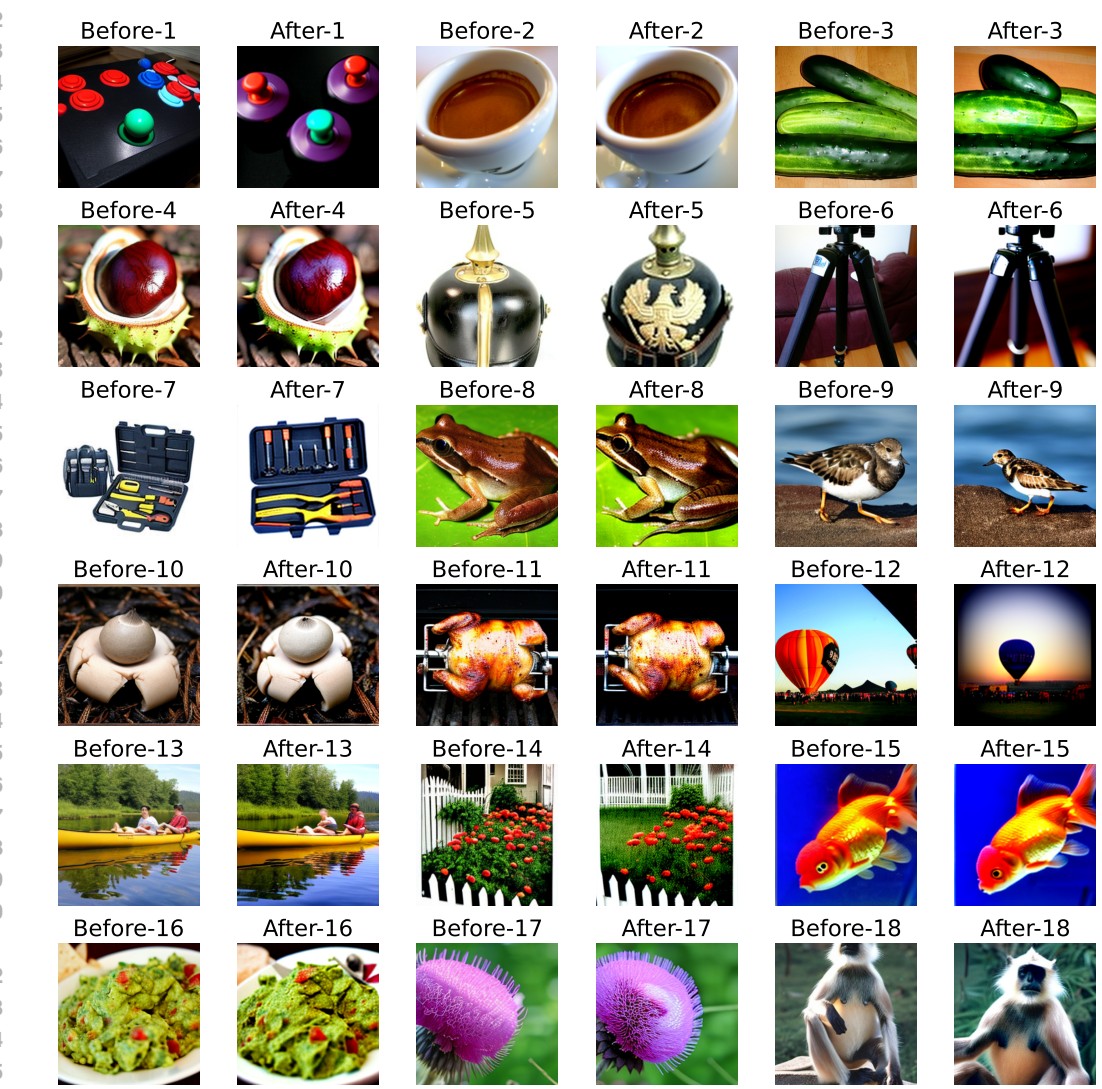

Figure 11: Results for unlearning on ImageNet with UNO-S, illustrated using 18 pairs of generated images. The images labeled "Before" were generated using the original model. Each image labeled "After" was generated after unlearning using the same noise sample for the diffusion transformer as the corresponding "Before" image.

We analyze now how a model, initially trained on the whole data set $\{X, Y\}$ with linear regression, changes during a single gradient decent step, and will find that the orthogonality term induces a gradient descent along the direction of largest variance of the retain data $X_r$ and a gradient ascent step along the direction of largest variance of the forget data $X_f$.

The cost function (3) for the linear model, as a function of the model parameters $W$, is written as

$$\mathcal{L}_{\text{UNO}}(W) = \|Y_r - WX_r\|^2 + \beta_o \| \left[ \nabla \|Y_r - WX_r\|^2 \right]^\top \left[ \nabla \|Y_f - WX_f\|^2 \right] \|^2. \quad (17)$$

The orthogonality term is readily evaluated as

$$\| \left[ \nabla \|Y_r - WX_r\|^2 \right]^\top \left[ \nabla \|Y_f - WX_f\|^2 \right] \|^2$$
$$= 16 \| \left[ X_r Y_r^\top - X_r X_r^\top W^\top \right]^\top \left[ X_f Y_f^\top - X_f X_f^\top W^\top \right] \|^2. \quad (18)$$

Table 3: Performance of various algorithms for class/feature unlearning with VAE on MNIST and CelebA when a classifier able to distinguish between the retain and forget data is available. Each experiment is repeated 10 times, and the standard deviations are shown in parentheses. Bold indicates the best score. ✗ indicates that the generated samples after unlearning are unrecognizably different from the original model. ✓ indicates the generated samples after unlearning are perceptually indistinguishable from the original model in terms of visual fidelity. An asterisk (*) indicates that, without classifier assistance, the algorithm failed to reach the target fraction of forget samples in the generated images within the allotted training steps.

| Dataset | Algorithm | Time to unlearn (s) ↓ | FID ↓ | Time per step (s) |
|---|---|---|---|---|
| MNIST (Class: 1) Original FID: 20.7 | H | 2.181 (0.853) | 23.4 (0.5) ✓ | 0.006 (0.0003) |
| | Ŝ | **0.021** (0.003) | 24.1 (0.5) ✓ | 0.008 (0.0006) |
| | UNÔ | 0.061 (0.009) | **22.1** (0.5) ✓ | 0.022 (0.0002) |
| | UNÔ-Ŝ | 0.041 (0.021) | **22.1** (0.4) ✓ | 0.017 (0.0067) |
| CelebA (Feature: Male) Original FID: 166.3 | H | 4.345 (1.849) | 174.8 (1.9) ✓ | 0.016 (0.0001) |
| | Ŝ | 0.181 (0.209) | 176.4 (3.5) ✓ | 0.023 (0.0002) * |
| | UNÔ | 0.499 (0.071) | **173.4** (1.8) ✓ | 0.178 (0.0010) |
| | UNÔ-Ŝ | **0.390** (0.218) | 175.2 (2.3) ✓ | 0.134 (0.0696) |

A gradient descent step starting from the model obtained from the whole data set $X$ is given by

$$W_1 = W_0 - \eta \nabla \mathcal{L}_{\text{UNO}}(W_0), \tag{19}$$

where

$$W_0 = YX^\top \left( XX^\top \right)^{-1} \tag{20}$$

is the least-square solution for the full data set. To simplify expressions we introduce the covariance matrices

$$\Phi_{r,f} = X_{r,f} X_{r,f}^\top \in \mathbb{R}^{d \times d}, \tag{21}$$

and the mismatch

$$E_{r,f} = Y_{r,f} - W_0 X_{r,f} \in \mathbb{R}^{1 \times N_{r,f}}, \tag{22}$$

and form

$$\theta_{r,f} = \nabla \left[ \|Y_{r,f} - W X_{r,f}\|^2 \right] = -X_{r,f} E_{r,f}^\top \in \mathbb{R}^{d \times 1}, \tag{23}$$

which we readily identify as the gradients of the standard unregularized loss function for the linear model restricted to the retain and forget data sets, respectively. Note that $\theta_r = \mathbf{g_r}$ and $\theta_f = \mathbf{g_f}$ (cf. (1)–(2)). Since $XY^\top = X_r Y_r^\top + X_f Y_f^\top$ and $XX^\top W_0^\top = (X_r X_r^\top + X_f X_f^\top) W_0^\top$, we have $\theta_r = -\theta_f$.

Introducing for simplicity of exposition $\beta = 64\eta\beta_0$, we obtain

$$W_1 = W_0 - \left[ \eta I + \beta \|\theta_r\|^2 \left[ \Phi_r - \Phi_f \right] \right] \theta_r, \tag{24}$$

which we can write as a two-step update

$$W_{\frac{1}{2}} = W_0 - \left[ \eta I + \beta \|\theta_r\|^2 \Phi_r \right] \theta_r \tag{25}$$

$$W_1 = W_{\frac{1}{2}} + \beta \|\theta_r\|^2 \Phi_f \theta_r. \tag{26}$$

Hence, for the linear model, the orthogonality term leads to a gradient descent step predominantly in the dominant eigendirection of the covariance matrix $\Phi_r$ of the retain data set $X_r$, followed by a gradient ascent step predominantly in the dominant eigendirection of the covariance matrix $\Phi_f$ the forget data set $X_f$.

