# OpenReview forum: "UNO: Unlearning via Orthogonalization in Generative models"
_ICLR.cc/2026/Conference — ICLR 2026 Conference Withdrawn Submission_

### Official Review · Reviewer_hirJ · 2025-10-29

**Soundness:** 3
**Presentation:** 3
**Contribution:** 2
**Rating:** 4
**Confidence:** 3

**Summary:**

This paper addresses the critical problem of selectively removing specific training data from generative models without costly retraining from scratch. The authors propose fast unlearning algorithms based on loss gradient orthogonalization for both unconditional and conditional generative models.

**Strengths:**

1. The paper is relatively simple and easy to follow.

2. The approach extends gradient surgery and multi-task optimization techniques to address known limitations (catastrophic forgetting,.etc) of simpler methods like gradient ascent and GDiff.

3.  The evaluation spans datasets of increasing complexity (MNIST, CelebA, ImageNet-1K) and different model architectures (VAEs, diffusion transformers), demonstrating generalizability and scalability.

**Weaknesses:**

1.  The approach builds heavily on existing gradient surgery methods from multi-task optimization; the paper primarily positions this as an application to unlearning rather than a fundamentally new algorithmic contribution. And it appears to be largely a combination of gradient descent on the retain set and gradient surgery on the forget set, lacking substantial algorithmic innovation beyond straightforward integration of existing techniques.

2. The paper only compares against relatively basic methods (gradient descent/ascent and gradient surgery), lacking comparisons with state-of-the-art unlearning techniques. It's difficult to assess whether the proposed method truly represents a significant advancement over the current state-of-the-art or merely an incremental improvement over older baselines.

3. The formalization focuses on preventing generation of similar samples but doesn't thoroughly address whether the unlearned information could still be extracted through adversarial prompting or other indirect means, which is crucial for privacy guarantees.

4. The visualization results demonstrate that the proposed method fails to effectively unlearn the targeted objects. Generated samples still contain recognizable features or instances of the supposedly forgotten data (eg. figure 3/4/5).

**Questions:**

1. A typo in Table 1, the smallest Time to unlearn of MNIST seems to be wrong?

---

### Official Review · Reviewer_fMLP · 2025-10-29

**Soundness:** 2
**Presentation:** 1
**Contribution:** 2
**Rating:** 2
**Confidence:** 4

**Summary:**

This paper proposes UNO (Unlearning via Orthogonalization), a framework for fast and selective machine unlearning in generative models. The core idea is to introduce an orthogonality constraint between gradients computed on the retain and forget datasets, enabling the model to forget specific data without full retraining. The authors introduce two main variants—UNO and UNO-S—and extend them to conditional and classifier-assisted scenarios. Experiments suggest that UNO achieves faster unlearning with comparable fidelity to gradient surgery while avoiding catastrophic forgetting.

**Strengths:**

- Presents a conceptually clear approach by introducing gradient orthogonalization into generative model unlearning.

- Includes algorithm pseudocode and reproducibility details, which enhance transparency.

- Provides demonstrations across several datasets and generative architectures.

**Weaknesses:**

Severely limited novelty. The orthogonalization-based formulation is conceptually similar to previously established gradient projection and subspace-based unlearning methods. Works such as "Machine Unlearning under Overparameterization" and the survey "Rethinking machine unlearning for large language models" already discuss orthogonal gradient and subspace decoupling techniques. UNO’s proposal appears to be an incremental adaptation rather than a novel framework.

Narrow experimental scope and inadequate evaluation. Experiments are limited to MNIST, CelebA, and a restricted ImageNet-1K setup. Proper evaluation of generative model unlearning should include large-scale, multimodal, and real-world benchmarks. The current paper fails to demonstrate generalization, scalability, or real deployment feasibility.

Lack of verifiable forgetting or theoretical guarantees. A key goal of unlearning is for the updated model to behave as if it were never trained on the forgotten data. However, the paper omits verification mechanisms such as membership inference, influence-function analysis, or retraining equivalence checks.

Parameter sensitivity and stability concerns. The performance of UNO depends critically on hyperparameters such as the orthogonalization weight and learning rate. No sensitivity or robustness analyses are provided.

Limited practical value and scalability. Despite claims of computational efficiency, the experiments only involve small-scale models. The method’s feasibility for large diffusion or language models remains untested.

**Questions:**

Refer to the Weakness above.

---

### Official Review · Reviewer_1fD5 · 2025-11-01

**Soundness:** 2
**Presentation:** 2
**Contribution:** 2
**Rating:** 4
**Confidence:** 4

**Summary:**

The paper proposed a new unlearning method by enforcing the loss gradient of retain and forget set be orthogonal, achieving utility preservation while erasing information of the forget set. The experimental results cover three dataset and two different generative models (VAE and diffusion), and achieved promising results.

**Strengths:**

1. The proposed method introduce a new regularization to further enforce the loss gradient of retain and forget data be orthogonal, to maintain the balance of knowledge removal and utility preservation.

2. The paper is easy to follow and understand, the proposed algorithm is clearly written.

**Weaknesses:**

1. The experiments are limited and there is no comparison to existing unlearning work cited in the related works.

2. The experiments only focus on class-level unlearning, no discussion on sample-based unlearning (or half sample in one classes).

**Questions:**

1. It is unclear to me that, for a conditional generative model (conditional on class), why we need a complicated method to unlearn, isn't it striaght-forward that just remap the class index to other classes before generation?

2.For both larger dataset experiment, it seems that the size of forget and retain set are similar, I wonder how the data imbalance affect the proposed method.
3. How does the proposed method work on classification task or text generation task? Based on my understanding of the algorithm, it does not restrict to any type of model.

---

### Official Review · Reviewer_2JhL · 2025-11-01

**Soundness:** 2
**Presentation:** 3
**Contribution:** 2
**Rating:** 4
**Confidence:** 4

**Summary:**

This paper proposes UNO (Unlearning via Orthogonalization) and UNO-S, two gradient-based unlearning algorithms for generative models. Both methods build on prior “gradient surgery” approaches by explicitly enforcing orthogonality between the loss gradients computed on retain and forget datasets. The authors claim that this orthogonalization prevents catastrophic forgetting while enabling faster unlearning. The paper evaluates the methods on VAEs and Diffusion Transformers, reporting improved speed and comparable FID relative to gradient surgery baselines.

**Strengths:**

- The algorithm is simple and easy to implement. UNO is just standard loss on retain data plus a cosine-style orthogonality penalty, and UNO-S alternates with surgery. Pseudocode and hyper-parameters are provided.

- Evaluations across different generative models and datasets demonstrate the general applicability of the framework.

- UNO and UNO-S achieve faster convergence in the “time-to-unlearn” metric while maintaining FID quality, indicating practical improvement over the gradient-surgery baseline.

**Weaknesses:**

- Limited technical novelty. The conceptual core, enforcing gradient orthogonality between retain and forget losses, is essentially a rephrasing of gradient surgery (Bae et al., 2023; Yu et al., 2020) and projection-residual unlearning (Cao et al., 2022). These works already formalize orthogonal or residual gradient projection to mitigate interference.

- Lack of theoretical contribution. The paper asserts that orthogonalization prevents catastrophic forgetting but provides no formal convergence analysis or generalization bound. In contrast, prior unlearning frameworks such as Neel et al., 2021 and Bu et al., 2024 derive theoretical conditions guaranteeing bounded deviation from retraining. UNO lacks such theoretical support, making its claimed stability largely empirical.

- Absence of core algorithm pseudocode in main text. The main algorithms (UNO and UNO-S) are only described verbally, while no corresponding pseudocode is presented in the main text (only referenced in the appendix). The omission of explicit pseudocode in the main body may cause confusion about implementation details and algorithmic distinctions.

**Questions:**

1. How does UNO’s regularization term differ fundamentally from the cosine-similarity loss minimization used in Normalized Gradient Difference (Bu et al., 2024) or PCGrad (Yu et al., 2020)?

2. Can the authors provide formal convergence guarantees or error bounds quantifying the deviation of UNO-updated parameters from a retrained model?

3. The Appendix lists ten algorithms (Algorithms 1–10), many of which appear to be variations or hybrids of each other. This abundance of algorithmic variants is confusing and potentially redundant. Are these methods derived through systematic combinations of a few base operations (e.g., orthogonalization + histogram loss), or do they each represent distinct contributions? More importantly, which algorithm should be regarded as the core contribution of the paper, and which are ablations or secondary variants?

---

### Note · Authors · 2025-11-20

I have read and agree with the venue's withdrawal policy on behalf of myself and my co-authors.